# Main Challenges for Child Digital Citizenship in a Consumer Culture in Brazil

Renata Tomaz [1,*] , Brenda Guedes [2,*] and Ingrid Martins [3,*]

1   School of Communication, Media and Information, Fundação Getulio Vargas, Rio de Janeiro 22250-900, Brazil
2   Post Graduation Program in Communication, Federal University of Ceará, Fortaleza 60440-900, Brazil
3   Facultad Latinoamericana de Ciencias Sociales, Buenos Aires C1050AAN, Argentina
*   Correspondence: renata.tomaz@fgv.br (R.T.); blguedes@gmail.com (B.G.); ingrid.seabra@gmail.com (I.M.)

**Abstract:** In this article, we reflect on how practices of children's consumer culture interfere with the exercise of rights by children who are consumers and producers of content on digital platforms. It is our aim to offer a communicational perspective to a broader discussion on the processes of child socialization within the scope of digital culture. So, this article intends to highlight some of the challenges for the exercise of children's digital citizenship based on the Brazilian experience. It also aims to insert Brazilian research in the international debate on children's rights on the internet. To carry out this discussion, we mobilize theoretical and empirical studies produced in Brazil and map national legal framework that supports the notion of digital citizenship for children. The theoretical effort of this work has pointed out at least two dynamics that explain the way in which the logics of consumption permeate the exercise of the rights of active children on social network platforms: the appropriation of the right to freedom of speech in order to enable child labor, and the conversion of the right to information into processes of publicizing brands in children's daily lives. We conclude that although Brazil offers a set of legal systems that guarantee the right of children to communication, the exercise of digital citizenship faces a series of challenges. In this sense, public policies which target children in the online ecosystem are needed so that they can participate in this environment without losing their protection guarantees.

**Keywords:** childhoods; digital rights; consumer culture; citizenship; kidfluencers

## 1. Introduction

### 1.1. The Aims of the Work

The relationship between media, childhood and technology has been investigated both from the perspective of opportunities for participation (Ponte 2020; Marôpo et al. 2018; Tomaz 2019; Sampaio et al. 2021) and from the perspective of constraints, such as those arising from dynamics of a consumer culture[1] (Buckingham and Tingstad 2010; Andrade and Castro 2020; Monteiro 2020; Zuboff 2020). In this article, we reflect on how practices of children's consumer culture[2] impose themselves on the exercise of rights by children who are consumers and producers of content on digital platforms. Our aim is to offer a communicational perspective of a broader discussion on the processes of child socialization within the scope of a digital culture. With this, the objective of this article is to give visibility to some of the challenges that arise for the exercise of child digital citizenship[3], and as a result, to stimulate the creation and development of public policies aimed at the best interest of the child in the online environment. In addition, we aim to include Brazilian research in the international debate on children's rights on the internet.

### 1.2. The Study Context

Talking about digital citizenship for children may seem obvious. However, if we remember that the internet started out as military technology, we realize that perhaps

younger people are more similar to foreigners than digital natives, a concept coined by Marc Prensky (2001) and quickly absorbed in the media, as well as in institutional and academic discourse. Around the second decade of the 2000s, however, this idea began to be problematized (Das and Beckett 2009; Koutropoulos 2011), especially in countries such as Brazil (Fantin 2016; Sampaio et al. 2017), where access to digital technologies replicates historic and profound social inequality. This type of inequality became even more explicit in the context of the Covid-19 pandemic, due to the intensification of demands for connection in the country.

The gradual and still growing occupation of the digital environment by children in the last 30 years is far from consolidated. In Brazil alone, in 2021, 2.1 million of the population aged between 9 and 17 years old lived in households without internet access (NIC.br 2022). The complexity of the Brazilian reality—similar to many other countries in the global south—shows the challenges regarding communicative practices that directly or indirectly affect children[4], ranging from the presence of artificial intelligence (A.I.) in media consumption experiences to the challenges of digital inclusion and access to technologies.

Broadly speaking, discussions on the rights of children who use social media mainly address the dangers and risks to which they are exposed when accessing these environments. Among the most discussed dangers and risks are bullying and sexual harassment (Barbosa et al. 2013; OCDE 2021). The guarantee of privacy has also been defended mainly in two aspects. One of them concerns the constant sharing of images of children by their parents, a phenomenon known as sharenting (Steinberg 2017; Blum-Ross and Livingstone 2017), which imposes an ethical dilemma around parental freedom of speech and the privacy of children. A second conflict concerns the collection and processing of children's personal data by applications and platforms for the purpose of personalized advertising (Henriques et al. 2021; Henriques 2021; ICO 2020; Montgomery et al. 2020). More recently, the debate on rights-by-design has increased (Hartung 2020), based on the idea that platforms need to offer engeneering and architectural solutions in development that address the digital presence of young people (Berriman 2013; Iversen et al. 2017; Vieira 2021).

In 2018, in France, the Parental and Digital Education Observatory initiated a mobilization to demand that youtuber children have the same legal treatment as those who exercise artistic activity[5]. In 2020, that country became the first in the world to have legal rules for digital influencers under the age of 16. In the United States, a lawsuit by more than 20 non-governmental organizations denounced YouTube to the Federal Trade Commission for collecting personal data from children, such as location and brand of cell phone used in access, to carry out targeted advertising[6]. In addition to paying a fine of $170 million in September 2019, the platform suspended personalized advertising from youtubers and disabled the option of users to comment on children's content in 2020.

In Brazil, the Public Prosecutor's Office in Rio de Janeiro launched a civil inquiry to "investigate possible violations of the rights of children and adolescents, arising from the dissemination of videos and images on the internet and possible characterization of artistic child labor" (Youtubers 2020, translated by the authors). In the same year, the toy manufacturer Mattel was sentenced to pay a fine of R$200,000[7] on charges of carrying out veiled advertising on the channel of a child youtuber. It was "the first case in Brazilian justice against advertising to children on YouTube" (Mattel 2020, translated by the authors). These examples inside and outside Brazil show how the debate on children's rights in the digital environment is taking place in the academic, governmental and civil society spheres.

In this way, the scenario shows how timely the launch of General Comment n. 25 (UN 2021) was, carried out by the United Nations Committee on the Rights of the Child. The document calls upon countries that are signatories of the Convention on the Rights of the Child (UN 1989), including Brazil, to develop public policies that allow children to exercise their rights on the internet, becoming, in fact, citizens in this environment.

Therefore, the perspective of digital citizenship adopted in this work is based on a humanist approach, situated in the field of communication, which problematizes relations between users and digital technologies and points to the need to empower these subjects

in more conscious, responsible, ethical and safe ways of using the internet. In this sense, some paradigms that occupy the public debate on the relationship between children, young people and digital media—such as digital security; risks and opportunities in online experiences; digital rights; digital well-being; as well as the technical and critical abilities required to make it possible to achieve beneficial and quality results in the experience—make up an important sphere of everyday life that also needs to be based on the principle of the best interest of the child, in force both in the UN Convention on the Rights of the Child (UN 1989), and in General Comment n.25 (UN 2021).

### 1.3. Methodological Procedures

This is a theoretical work of an exploratory nature. To carry it out, we first set out a series of theoretical and empirical investigations, mainly in the Brazilian context of the 21st century, which offered a list of consumption practices engendered in children's modes of digital sociability. Since our objective was to find the most recurrent debates, in the Brazilian context, on the relationship between children, consumer culture and the internet, our strategy was to collect these publications through a Google Scholar search using keywords and applied four filters from the platform: period (2000 to 2022), language (Portuguese), type of document (any) and ordering (by relevance). As the sorting filter allowed us to rank the works by relevance, it was useful to identify the most recurrent subjects within the search terms.

From the total results presented by the search engine, we discarded those that did not originate from the field of communication. The themes identified in this sample touched mainly on four debates: (1) children's exposure to advertising in the digital environment; (2) their emergence as content producers, mainly on YouTube; (3) the consumption they carry out in digital media; and (4) the construction of youth representations in online spaces. Considering that these themes come from works published by researchers, research programs and journals in the field of communication, in Brazil, this sample limits us to the debate of just one specific field. In addition, the choice for pages in Portuguese left out Brazilian communication studies that have been published in foreign languages.

In parallel, we mapped treaties, laws and resolutions that support the notion of the rights of Brazilian children in the digital environment in order to problematize how the online childhood experiences and the exercise of the rights linked to them, particularly in communication, incorporate market logics stemming. To discuss how these logics impact children's interactions—with regards to children's rights—we chose to reflect on two of the four debates identified in the initial sample: marketing communication targeted at children and child labor. Such practices will be described and discussed from a communicational perspective, in dialogue with the sociology of childhood and the field of investigations on digital rights. The reflection showed that although there is a legal interpretation that supports the rights of children in the digital environment, it is necessary to develop specific public policies to enable them to enjoy broadly diverse opportunities.

The text is divided into three main parts. The first part presents theoretical grounding from which we approach children's consumer cultures. Next, we focus on the legal provisions that deal with fundamental principles for the understanding of the rights of Brazilian children as communicational subjects. In the end, we present two types of dynamics we identified in the analyzed studies, which explain how the logics of consumption permeate the exercise of rights by active children on social media platforms.

## 2. Communication and Children's Consumer Cultures

Children's relationship with the internet exposes elements of the unmistakable connection between the social construction of childhood and communicational processes, perceived and studied by different perspectives in the human and social sciences. Phillippe Arriès ([1960] 1981) identified the formation of a distinct feeling of childhood in the pictorial representations of children in the French *Ancien Régime*. Joshua Meyrowitz (1985) stated that television undid the secrets that supported the boundaries between the adult and

child universe, leading younger people to question age-based social hierarchies. For David Buckingham (2007), the use of media by children reveals their ability to produce meaning and re-signify realities. Livingstone (2011) argues that societies can create safer and more friendly digital environments for children when they are aware of how they use digital technologies.

The examples above show the communicational processes crisscrossing the social construction of childhood in different historical contexts. They also illustrate the role of communicative practices in the processes of socialization (Tomaz 2020). At the same time, these different theoretical perspectives reveal political challenges for the social insertion of children. On the one hand, the languages of the media release children from a long traditional kind of literacy so that they are enabled as interlocutors of the societies in which they are inserted. On the other hand, as a result, they are exposed to different actors who challenge them in different ways. This means that, when using media cultures to achieve more visibility for their generational demands, these young users are also more called upon to respond to all kinds of urgencies, including marketing.

Commercial discourse from brands and companies addresses children directly and/or indirectly as they collaborate to reproduce practices and rhetoric aligned with a neoliberal project[8] for society. In this sense, both children's expressiveness and representations potentially respond to a type of social reality that is deeply mediated by media communication that predominantly serves a specific model of culture: the consumer culture. Steinberg and Kincheloe (2011) call "kinderculture—the corporate construction of childhood" the way these discourses engender dialogues with children, primarily as consumers.

Such an understanding is in line with what Daniel Cook (2004, p. 12) points out as a process of commoditization of childhood itself. Here, part of the legitimation of children as social subjects starts with their recognition as consumers.

> Indeed, children's "right" to consume in many ways precedes and prefigures other, legally constituted rights. Children had been given a "voice" on the retail sales floor, in "design-it and name-it" contests, in clothing choice, and in marketers' research designs decades before their rights were asserted in such contexts as the UN Convention on the Rights of the Child in 1989. Children's participation in the world of goods as actors, as persons with desire, underpins their current, emergent status as rights-bearing individuals.

The convergence between traditional and new media in the contemporary context of a so-called participatory culture (Jenkins 2009) also proposes the figure of the media prosumer (Toffler 1980), who articulates consumption and production of content together with communication platforms. Someone who consumes and produces information, assuming a leading role, as an example of what is posed as a reality for kidfluencers (Callens 2020; Masterson 2021; Feller and Burroughs 2021), children with increasingly notability and a growing audience on digital platforms. The sheer number of followers of some of these children, as evidenced later in the work, causes them to be contacted by advertisers interested in promoting their brands in the content posted. Thus, both the consumption and the media production of the youngest can function as vectors of the logic of commoditization that, according to Zygmunt Bauman (2008), concern the transformation of everything—including the self and the other—into merchandise.

## 3. From Law to Guiding Principles: Guarantees for Children as Subjects of Rights

In Brazil, any perspective of regulating communicative practices regarding children should strengthen the legitimacy of children as social subjects of rights. They also need to be defined by legal protection instruments, consolidated from the establishment of a democratic regime in Brazil. Thus, the precepts established, for example, by the Federal Constitution of 1988 (Brasil 1988), by the ECA—Estatuto da Criança e do Adolescente (Statute of Children and Adolescents) (Brasil 1990a), by the CDC—Código de Defesa do Consumidor (Consumer Defense Code) (Brasil 1990b), by the Marco Civil da Internet (Civil Rights Framework for the Internet) (Brasil 2014) and by LGPD—Lei Geral de Proteção

de Dados Pessoais (General Law on Data Protection) (Brasil 2018) are fundamental legal mechanisms for understanding the rights of Brazilian children.

More than recovering in full the various regulatory instruments—which in practice are connected in complementarity mode—we chose to build an argument for this topic of our reflection that is based on three ethical guiding principles. These will be activated and reinforced from the regulations and legislations themselves. They are the absolute priority of the child, even if it meansto the detriment of other social segments; the consolidation of an understanding of children as subjects with the right to communication; and the special protection that this public enjoys, under the terms of the law, in the face of situations that characterize the exploitation of their vulnerabilities.

The absolute priority statute, bestowed upon Brazilian children, appears explicitly in article 227 of the Federal Constitution (Brasil 1988) and subsidizes the ECA (Brasil 1990a) in defining the fundamental rights of this group, for which it defines a series of guarantees. A look at younger people as subjects in direct dialogue with socio-communicational aspects is also drafted from specific points through which the ECA (Brasil 1990a) covers cases of protection of "minors" in the mass media; details legal penalties for cases of child pornography in theatrical, cinematographic, television, photographic or any other visual medium and also specifies penalties for similar cases on the internet.

The Consumer Defense Code (Brasil 1990b) connects to the debate insofar as it presents itself as a legal resource available for observing marketing communication practices. The document has three articles that deal with advertising. One of them clearly shows what is considered as misleading advertising and abusive advertising. Regarding the latter, as indicated below, the text of the law directly addresses the situation regarding children.

> Art. 37—§ 2° It is abusive, among others, discriminatory advertising of any nature, which incites violence, exploits fear or superstition, takes advantage of the child's lack of judgment and experience, disrespects environmental values, or can induce the consumer to behave in a manner that is harmful or dangerous to their health or safety. (Brasil 1990b emphasis added)

In a context of social pressure for companies to adopt an ethical standard when advertising their products and services, in March 2014, the National Council for the Rights of Children and Adolescents (Conanda 2014) approved Resolution 163, that deals with "the abusiveness of targeting advertising and marketing communication to children and adolescents" (Conanda 2014, translated by the authors). This Resolution lists several aspects that can configure abusiveness. For Craveiro and Bragaglia (2017, pp. 72–73), the measure represents an effort that aims to "draw an overview of the elements of advertising language that are more receptive and, consequently, more persuasive to child audiences". On the one hand, civil society movements that work in the field of childhood rights perceive Resolution 163 as a "complement of the microsystem for the protection of collective rights" (Thibau and Rodrigues 2015) and, therefore, a great advance in legislation. On the other hand, business sectors question the legitimacy of such a measure because it is not a law approved by the National Congress.

In 2014, the institution of the Civil Rights Framework for the Internet (Brasil 2014) also ensured rights and guarantees for internet users that also extend to children. It matters to our debate insofar as it links access to the network to the specific possibilities of exercising citizenship which, in turn, must be preceded by rights such as the inviolability of intimacy and private life; protection and compensation for material or moral damage resulting from such violation; and the requirement for clear and complete information on collecting, use, storage, processing and protection of personal data—which may only be used for specific purposes.

The intensification of the processes of digitalization and datafication of life forcefully targets children who, according to Unicef (2017), represent a third of internet users on the planet. It can be said that the perspective addressed by José Van Dijck (2017), on the importance of paying attention to the transformation of social action into quantified online data, which results in real-time monitoring and predictive analysis, was contemplated

in Brazil by the proposition of the General Personal Data Protection Law—LGPD (Brasil 2018). In Article 14, the law deals with the interaction of children in the digital environment, stipulating the best interest of the child audience as a basic ethical principle for the performance of activities such as the processing of personal data—a parameter that refers to the perspective defended by Brazilian researchers and activists about rights-by-design (Hartung 2020; Instituto Alana and Internetlab 2020).

Other parameters are articulated to reinforce this same line of argument. They are based on multilateral conventions and treaties that support the need for democratic regulation of what is practiced in the various media (and other spaces that include the presence of children) and demand commitments and provision of accountability services on the part of the signatory States. The UN Convention on the Rights of the Child (UN 1989) is one of the most important subsidy instruments in this area. As a State Party to the Convention, Brazil guarantees the fundamental rights of all people under the age of 18, including freedom of expression and the broad perspective of the right to communication, which permeates practices that intersect children and commercial discourses.

The recent General Comment n. 25 (UN 2021) on children's rights in the digital environment deals, among other things, with how, from the children's own point of view, digital technology is vital for their future, and how they believe that this environment should support, promote and protect their engagement in a safe and equitable manner. This document was prepared by listening to more than 700 children and young people between the ages of 9 and 22, from twenty-eight countries on six continents. It is based on the general principles of non-discrimination, the best interests of the child, the right to life, survival and development, and respect for the child's opinion, indicating "the measures needed to guarantee the realization of children's rights in relation to the digital environment" (UN 2021, p. 2). Among the measures, we highlight the attribution of responsibilities concerning the relationship between the "children's rights and the business sector" and the attention to issues that concern "commercial advertising and marketing".

In addition to the legal instruments mentioned, the code of the Advertising Self-Regulation Council (Conar 2013) condemns practices commonly used by digital influencers such as merchandising or indirect advertising that uses children. It advises, in Article 37, that children should not be used as models for direct appeal, recommendation or suggestion of consumption of a product or service. Furthermore, in the absence of current legislation that specifically deals with children as producers of commercial content on digital platforms, Conar has published the *Advertising Guide for Digital Influencers* (Conar 2021), with guidelines for the application of rules from the council's own code to commercial content on social networks, with a focus on user-generated content by so-called "influencers".

All of the legislation referred to here is based on the inescapable premise of children as subjects of rights whose identities are also constituted amid communicational consumption processes. In this sense, as a right, communication is related to the possibility of dialogue and social participation, insofar as it has the potential to ensure the guarantee of other human rights and the construction of a more democratic society. The processes of construction and reaffirmation of citizen identities with children therefore demand co-responsibility between family, society (including companies such as digital platforms that operate in Brazil) and the State not only in the application of current legislation but, mainly, in the adoption of broader ethical principles consistent with the experiences of children who occupy the digital scene.

## 4. From Exercising One's Right to the Violation of Rights: Tensions in Children's Digital Citizenship

After mapping Brazilian legal systems that support the comprehension of children's rights in the digital environment, we analyzed theoretical and empirical studies carried out in the Brazilian context of the 21st century on children's consumer culture practices in digital modes of sociability. We seek to answer the research question on how the enjoyment of these rights by children is impacted by the marketing interests that guide the use of the

internet by younger generations. In this topic, we discuss the results of this theoretical effort, presenting at least two dynamics that explain the ways in which the logic of consumption permeates the exercise of rights by active children on social media platforms. First, by transforming cultural production into a commodity and, thus, creating loopholes for child labor and, second, by inserting brands and products into the daily narratives of children who use them to address marketing communication to their peers.

In this sense, this topic addresses consequences arising from the development of children's identity models of citizenship amalgamated with consumer culture. We start from the premise that, as communicative subjects, children perform multiple interlocutions, becoming the target of several interpellations. With this, we argue that, as soon as children are constituted as social actors, they assimilate the operating logics of these spaces that allow for public communication within a corporate domain. It is these logics—arising from private interests—that end up hurting and violating other rights guaranteed to the youngest individuals.

### 4.1. From Freedom of Expression to Child Labor

The content made for, with and by children on the internet is in greater volume in the audiovisual format. These are videos about how they play, who they play with, what and where. They can also include tutorials dealing with culinary, artistic, media skills; travel vlogs and celebrative dates such as birthdays and Christmas; and tours of a new house or a newly decorated room can also motivate children to post videos online. They also produce doll novels, theaters and web series. Challenges and competitions with family and friends, sharing intimate moments, preferences of all kinds and moods are also part of their content. So is the exhibition of consumer goods, success stories and trollings, also known as "gotchas". This variety of subjects and frameworks exemplify some possibilities for children's cultural participation in connected societies.

The production of children's content in the digital environment can be thought of as a cultural production, as it expresses the meanings arising from interactions between children. For William Corsaro (2011), the results from interactions among peers—that is, their perceptions, learning, games, rules and narratives—form children's cultures. Manuel Sarmento (2003) identifies both an intra- and an inter-generational character of children's cultures, because he understands them as convergence. According to the Portuguese sociologist, what is produced and offered to children—whether in the domestic, school or media environment, for example—converges to their interactions and, through a synthesis, supports processes that can question or reproduce realities. Thus, what children are publishing—or what they are watching—on digital platforms can reveal, on the one hand, that they develop creative ways of reading and narrating the world. On the other hand, this production can also reveal that they operate mechanisms that reproduce existing social constraints.

In this scenario, the media phenomenon of kidfluencers, producers of content for different digital platforms that thematize an increasing number of studies in Brazil, is quite emblematic (Andrade and Castro 2020; Corrêa 2016; Marôpo et al. 2018; Monteiro 2019; Monteiro 2020; Tomaz 2019). These are surveys that recognize the creative activity of children on the network, who have gained notability for scriptwriting their ideas, recording, editing and publishing diverse content, with the collaboration of their guardians and peer groups. Initially, their activities drew attention for being able to express, among other aspects, learned communicative skills, child protagonism, social recognition and identity projections. However, a strong point for the kidfluencers' recognition is linked to the volume of followers they accumulate and the financial gain from the monetization of their contents.

Brazilian digital influencer Julia Silva, for example, started posting videos of toys that she and her mother made or toys that they received when she was 6 years old, in 2011. She was one of the first children in the country to reach the milestone of one million subscribers on YouTube. For this reason, she has become a reference for children's

audiovisual production. Her channel is among the most analyzed in the Brazilian context (Brum and Schimidt 2016; Marôpo et al. 2018; Miranda and Sampaio 2020; Monteiro 2019; Tomaz 2017). The channel was also a case study in ethnographic research on children's digital playful culture (Tomaz 2019). Videos, such as those by Julia Silva, not only showed toys but also possible ways of approaching them, handling them, granting them new meanings. These videos presented narratives and built different environments by which they could transcend traditional ways of playing. Marôpo et al. (2018) took this and other YouTube channels as an object of investigation to analyze the social participation of children. They noticed some socializing dynamics, including the use of communicative strategies to give visibility to the construction of female youth identity. Moreover, though both works pointed to the creative way in which the so-called youtubers exercised freedom of expression by means of communicative processes, they also pointed out the clear corporate mediation of this right.

Engendered in the interactions carried out in the digital environment, meaning-producing actions, knowledge and visibility of children carry great mobilizing potential among their peers. Kidfluencers' publications generate many touchpoints on social networks, as they gather millions of followers. The protagonist of the Ryan's World channel, for example, topped the list of the highest paid youtubers in the world, published annually by Forbes[9] magazine, in the years 2018, 2019 and 2020, ranking seventh in 2021. At the age of 10, Ryan Kaji surpassed the 30 million subscriber milestone in 2022, with about 48 billion views. In Brazil, there are also children's channels with strong engagement, such as Valentina Pontes (22.6 million followers), and Maria Clara and JP (32.8 million followers)[10].

As soon as children can mobilize a large network of followers, they gain social as well as economic recognition due to their ability to influence behavior. The offer of sponsorships, gifts, contracts and invitations to advertising campaigns directly affects not only the content itself, but also its nature. Their videos then move from cultural production to a (potentially) commercial production (Gorshkova et al. 2020). Although it is no longer possible, since April 2020, to monetize children's videos on YouTube, content producers can publish sponsored material, which allows for the commercial exploitation of children's content. Tomaz (2019) approached this change in nature from the perspective of a shift from playful culture to playful capital, as children's content producers realize that they can transform social recognition among their peers into individual monetary gains.

Marôpo et al. (2018, p. 192) approached this movement from the point of view of entrepreneurship of the self. In their words, the self is then "subordinated to the logics of consumer culture and managed in an increasingly professional way". The recording routine, commercial commitments, fierce competition for visibility on different platforms, the effort to influence other children and the assimilation of a repertoire of consumption practices (Barcellos 2020; Primo et al. 2021; Quintian 2018) converge to what is increasingly being outlined as a professional activity performed by children on social networking sites (Dantas and Godoy 2016)—a clear violation of rights guaranteed to persons under 18 years of age both nationally (Brasil 1988, 1990a) and internationally (ILO 1973).

If the field of communication in Brazil, albeit recently, has advanced with research that exposes the functioning of children's content production on digital platforms by children, Brazilian law research stands out for the most purposeful works in terms of regulation (Andrade 2022; Frazão 2021; Zanatta et al. 2021). Considering the illegality of child labor, both in the Federal Constitution (Brasil 1988) and in the Statute of Children and Adolescents (Brasil 1990a), the productions demand labelling children's commercial content creation activities as artistic child labor.

This argument is based on the idea of guaranteeing the right of expression and, at the same time, the enjoyment of legal protection regarding working and studying time, remuneration and physical and emotional protection mechanisms (Medeiros Neto and Marques 2013, p. 41). This was the perception of the French government when it approved law that guarantees children who produce content for the internet the same rights as those

who perform artistic child labor[11], an exceptional type of work provided for in Convention 138 of the International Labor Organization—ILO 1973), also in effect in Brazil[12]. Along with this effort, research has begun to emerge that discuss the regulation of these activities by means of specific legislation (Andrade 2022; Vargas 2022).

If, on the one hand, the activity of content producer is leading some children to professional practice in an unquestionable violation of their right to not work, on the other hand, preventing such activity is also depriving them of the right to self-expression. In this sense, the debate shows us that there must be a regulation in which the rights to participation and protection, guaranteed by the 1989 Convention, are not in opposition, but in conjunction with the online experiences of children, as advocated by the General Comment n. 25 (UN 2021). From our point of view, this starts with—but is not limited to—recognizing artistic child labor in order to offer proper protection to those who exercise it. In addition, we understand the need for a public policy that considers the participatory insertion of children on the internet, not only from a legal perspective, but especially from an educational perspective. In other words, a digital literacy policy that establishes principles for exercising citizenship in the online context.

### 4.2. From the Right to Information to Publicity Strategies in Children's Daily Lives

Nowadays, what children consume in terms of media points to the condition of children's mediatic prosumers (Toffler 1980)—children who relate to communication content both from the perspective of consumption and production. The internet has become a place where children talk about themselves, converse, exchange information, play, study, learn and, consequently, experience different forms of sociability (Andrade 2020; Tomaz 2019). However, while they are exercising their right to information and communication (Brasil 1990a; UN 1989), they are exposed to interlocutions that, in different cases, violate other aspects protected as a focus on their best interest. It can be said, therefore, that although they expand their communicative experiences, they also incorporate multiple dynamics that organize digital experiences, among which are those governed by commercial logics.

The contact of Brazilians aged 10 to 17 with advertising on video sites rose from 43% in 2014 to 67% in 2018 (CGI.br 2021). In 2020, 62% of individuals in this age group claimed to have had contact with videos of people opening a product package (unboxing); 58% had contact with videos of people teaching how to use a product, while 52% with videos of people showing products received from a given brand; 47% admitted having had contact with videos of people going to stores or events to show a product or brand; 47% with videos of people making challenges or games with a product or brand; and 45% with videos of sweepstakes or contests for products or brands (CGI.br 2021). These figures reveal the pervasiveness of advertising discourses in the content consumed by younger internet users.

By being more present in the online environment, children also become more visible, attracting different companies, brands, and manufacturers. This insertion in entertainment content meets the basic rule of advertising to convey messages in spaces where the target audience is (Lewis 2001). Furthermore, the digitalization of everyday life processes, making individuals more and more web tracked, by having their data collected and stored. This factor makes it increasingly easier for advertisers and advertising agencies to find children, especially on social media. These are spaces that, despite a growing presence of children, are slow to offer an interface that is not only friendly, but also architecturally safe for this segment of the population. Thus, the increase in children's occupation in the digital world has also driven the use of different strategies to target children.

It is possible that some view the internet as a more protected environment from commercial discourse than vehicles such as television. There are no commercial breaks demarcated in blocks with a sequence of advertisements and there is the possibility of skipping the advertisements, for example, in the display of some types of videos. However, the digital medium has its own formulas for displaying advertising content, whether through more traditional formats specific to the environment, such as banners and pop-ups; advertising strategies for brands, products and services that are hybridized with the many

contents present on the network (Martins 2020); or the collection and processing of personal data for the production of personalized advertising—a practice condemned by the General Comment n. 25 (UN 2021) and which led to YouTube being punished in 2019[13].

In Brazil, the debate on the relationship between advertising and childhood comprises both the argument of the right to consumption (ABAP 2013) and the right to protection against marketing communication (Alana 2013). The first group includes Conar (National Council for Advertising Self-Regulation), ABAP (Brazilian Association of Advertising Agencies), ABA (Brazilian Association of Advertisers) and Abert (Brazilian Association of Radio and Television Broadcasters). Aligned with the second group are entities such as Recria (Research Network on Communication, Childhood and Adolescence), Instituto Alana (Alana Institute) and Andi (Child Rights News Agency). Researchers, legislators, jurists, families, and non-governmental organizations participate in this discussion, putting into circulation different interpretations on the asymmetry of power in children's relations with market discourses. For some, practicing child advertising is not a problem if there is a clear and evident identification of the practice (Andrade 2022). For others, Conanda's Resolution 163 displays the challenges that justify the prohibition of all marketing communication aimed at children (Bragaglia 2013; Sampaio 2009; Martins 2020), both in offline and online contexts.

In addition to the discussion about the illegality of what is conventionally called advertising for children, another much debated topic involves the forms of presentation of advertising discourse. As marketing communication is increasingly mixed with other available contents on the internet, identifying which messages are of a commercial nature becomes a complex activity, especially for the younger audience. According to Maria Clara Monteiro (2020), there is a growing use of marketing strategies disguised as content in the digital environment, such as on social media platforms. She investigated how a group of elementary school students related to advertising in videos by Brazilian youtubers and noticed that, although children show impatience and dissatisfaction with the advertising formats standardized by YouTube (banners, display ads, masthead etc.), they tend to have low resistance to the presence of brands and products in influencers' content. By classifying this insertion in the channels that she analyzed, Monteiro (2020) identified a series of video genres that indicate the potential hybridization of entertainment with advertising, for example tours (around the youtuber's house or room), shopping (in stores or websites), challenges (using soda brands, chocolates, make up etc.), unboxing (opening packages), collections (objects of different nature) and promotion (youtuber's own brand dissemination). Each of these possibilities amplifies the reach of brand publicity strategies (Casaqui 2011).

It is necessary to add to these practices the fact that, in 2019, videos from digital influencers were the third most consumed content on the internet by Brazilians between 10 and 17 years old. In most cases, these videos contained the presentation of products and brands (CGI.br 2021). As defined by Peres and Trindade (2017, p. 2), youtubers are "opinion-forming agents", which gives them the authority to suggest products, services, and brands. Social media is among the most widely chosen space for these dynamics. In 2020, WhatsApp (86%) and Facebook (61%) were the platforms with the highest number of profiles of individuals aged between 10 and 17 in Brazil. However, Instagram (35%) and TikTok (27%) have emerged as the most widely used platforms by this age group (CGI.br 2021).

The Instagram profile of Pietra Quintela[14], a 14-year-old actress with 6.5 million followers, is an example of this dynamic. In her posts, Pietra often tags in her photos the brands she is using, not always mentioning whether this is paid content. Such an indication is commonly manifested with the hashtag #publi. In a publication made in July 2022, for example, the actress posted a sequence of photos with a hair treatment product, tagging the brand in the caption with the following text: "Tip for those who suffer from frizz in their hair like me: Frizz Zero by Truss, instant effect and practicality with three possibilities of disciplining the hair strands. ( . . . ) And @trusshair has a promotion on the website, buying

Frizz Zero you get a travel size version of it. Unmissable huh?!". In such cases, the child persona provides the opportunity to adhere to marketing content manifested by various brands throughout the posts.

Some child digital influencers have accounts and channels on different platforms. This can optimize the potential reach of their actions, in terms of resources. More than that, the contents produced and published in these spaces are mixed by means of crossposting strategies. In this sense, in addition to Instagram and YouTube, TikTok has presented itself as yet another platform that, nowadays, accommodates diverse interlocutions with children (Kennedy 2020). In fact, it has risen to the status of "a sensational app of the moment" (Kennedy 2020, p. 19) in the pandemics. Their videos with trendy dances have had repercussions in the press as "the perfect antidote against child and adolescent boredom resulting from the quarantine" (Lopes 2021, p. 173).

It can be said that brands insert themselves into children's daily lives by means of marketing dynamics that shape and adapt to the entertainment experiences characteristic of connected childhoods (Andrade and Castro 2020), as discussed above. This is often carried out without contemplating the principle of advertising identification—required under Brazilian law dealing with consumer protection (Brasil 1990b) and recommended, in addition, by the country's self-regulatory organizations. In this way, discussions about the need to protect children from such attacks gradually gain strength (Martins 2020; UN 2021; Lopes 2021). After all, even though social media has become a space for children's participation and, in this sense, for the exercise of rights (Sampaio et al. 2021), they also serve as the propagation of marketing messages, reinforcing a neoliberal project of society integration on contemporary children's playful cultures.

By promoting connections between their brands and the channels and pages starring and/or consumed by children, advertisers violate the basic ethical principles for prioritizing the child subject in social relationships. According to this same line of reasoning, initiatives with an advertising bias directed at this audience also in digital environments can be framed as misleading (Brasil 1990b) considering, for instance, the concealment of sponsored content and/or the existence of advertising contracts. When the public consumes content without knowing that such content is part of marketing discourse, their right to information is violated. Such right is not to be confused with the fallacy of the right to consumption (Aquino and Gomes 2010).

In the contemporary Brazilian scenario, the challenge of instrumentalization of law to deal with the phenomenon addressed in this article has resulted in a dispute of narratives (Guedes 2016). This also points to how, in the eyes of advertisers, there are no legal infractions incurring in the practices described here. However, as highlighted by Rizzatto Nunes (2009, p. 478), it is not necessary that actual damage or offense to the consumer takes place: "it is enough that there is danger; that there is a possibility of damage, a violation or offense. Abusiveness, by the way, must always be evaluated in view of the potential of the advertisement to cause harm".

Although there is a legal understanding about the abuse of marketing communication targeted at children in Brazil, this work reveals that the online environment does not yet seem to be understood—by some social actors—as a space for exercising rights such as the offline does. In this sense, it is vital to consider the General Comment n. 25 (UN 2021), with respect to the formulation of public policies by the Signatory States, and to create mechanisms that educate not only children, but society in general, to think about the protection of children on the Internet as necessary as in any other environment. As children need to have their rights recognized and respected, it is necessary public policies that, based on the ethical principles mentioned throughout this work, consider the specificities of commercial interpellations which target children who are on the Internet.

### 4.3. Potentialities and Limitations

By consuming and producing content in different formats (audiovisual, imagery, sound, text) and of different natures (playful, educational, informative, instructive), children

are accessing not only an online environment, but a range of social, economic, political and cultural realities. So, in such experiences, they are socializing and being socialized, building identities, playing different roles (Cavalcante and Sampaio 2020; Jiménez et al. 2016; Othon and Coelho 2020; Dias 2020). However, the practices that insert them into these dynamics are forged in a sphere that, even being public communication, is predominantly corporate, which is why its operation is governed by private interests. Thus, quickly and easily, cultural or recreational content becomes commercial while the child that watches and is entertained in that environment is addressed as a consumer, and the child who is a producer is addressed as a sales agent. In this sense, it is possible to identify a tension between the spontaneous and legitimate uses children make of digital media and the marketing practices that support the business models of the platforms. The present work approached this tension.

The discussion carried out on the two dynamics that involve children's sociality on the internet, in the context of a consumer culture, presents, on the one hand, some potentialities and, on the other hand, some limitations on at least three levels: academic, social and political.

In academic terms, even though children's rights on the internet are increasingly being discussed in scientific publications, the literature points to a gap in this production regarding the experiences of the global south (Livingstone and Bulger 2014; Nawailaa et al. 2018). This work strives to respond to this demand, as it inserts Brazilian studies into an international scenario, showing local specificities of a phenomenon that is global. Considering the initial nature of this work, such an effort comes up against some limitations imposed by the methodological choices. The topics chosen for discussion, for example, do not originate from a systematic literature review, but from a group of publications identified by Google Scholar, between 2000 and 2022, in pages in Portuguese, from the field of communication. Topics such as children's data protection and the practice of sharenting, and areas such as education and psychology, were left out. To advance from this point, in addition to a state of the art that addresses these gaps, future research on the topic discussed here could propose comparative investigations between Brazil and other contexts from both the North and the Global South.

The social potential of the work lies in the fact that it makes explicit the mechanisms through which children's digital sociability incorporates the commercial logics established by the platforms where they occur. Putting this research into dialogue allowed for the identification of a gradual increase in the touching points between commercial initiatives and children, making them the target of brands and manufacturers, whether they are consuming or producing content on the internet. It shows, therefore, how the social role of children adapts to market demands, particularly in the digital environment. This work, however, does not describe or analyze such mechanisms, operating through strategies such as the YouTube partnership program or through usage policies, such as content production guidelines. In this sense, it is not only opportune but also necessary to carry out investigations that focus on the platforms and their ways of enabling and favoring that different actors in the market address and relate to children in their different dynamics.

Politically, this work fuels a growing discussion about the exercise of rights by children in the online environment. In this sense, it is part of what Livingstone et al. (2018) called the third era of research on the use of the internet by children. It is a phase marked by investigations concerned with the formulation of public policies, "[ . . . ] to provide a firm but critical foundation for a forward-looking policy and the practice needed to keep pace with a fast-changing socio-technological environment." (Livingstone et al. 2018, p. 3). The demand for policies that ensure the protection of children in their online experiences indicates the strengthening of the notion of children's digital citizenship. This means that, in addition to being consumers and producers of online content, they need to be increasingly understood, in these spaces, from their citizen identities, which must be constructed based on facing the violation of their conferred rights (Livingstone et al. 2020; Monteiro et al. 2020; Sampaio et al. 2021).

We recognize, however, that the formulation of public policies based on these notions requires not only a discussion but also a more robust conceptualization of children's digital citizenship. A conceptualization that includes aspects such as the role of education, an element not explored in this work, but central, in our view, for younger generations to carry out their participation in the online environment with the necessary protection. This interlocution may be resumed in due course in works that propose to advance this point.

## 5. Conclusions

Although Brazil offers a set of legal systems that guarantee the right of children to communication, the exercise of digital citizenship faces a series of challenges. Consumption and production of media content by children on digital networks broaden their interlocutions, leading them to be challenged by different types of discourse, including commercials. On the one hand, their interactions are creating multiple meanings. On the other hand, they are providing them with a repertoire of narratives that structure practices, dynamics and identities both online and offline. Therefore, in this work, we problematize children's interactions mediated by digital technologies to identify how the logics of consumption crisscross the rights of children as communication subjects in connected societies.

One of the arguments developed here is that the agency conferred to children by consumer culture in the environments of digital social networks, is strongly structured by neoliberal discourses, that is, the narratives and practices that express the values of a neoliberal society. These are guided by the individualization and self-responsibilization of child actors. Based on theoretical and exploratory research anchored in the Brazilian context, we discuss two dynamics through which contents made by, with and for children reveal the shift from the exercise of the rights to its violation. On the one hand, children's cultural production is commoditized, turning the youngest into sales agents and inserting them into a labor practice. On the other hand, the contents aimed at them present narratives that incorporate and, thus, naturalize the presence of brands, products, and services in children's daily lives. For this reason, we defend the perspective that the marketing challenge to children is, under the terms of the law, an abusive practice and we recognize the need for an approach anchored in ethical principles that guide the formulation of public policies that respond to the questions to which we give visibility here.

This discussion led us to the understanding that the existence of laws that support the participation of children in the online environment does not prevent the violation of some of the rights guaranteed to them. Secondly, it proved to be fundamental to elaborate public policies that allow for the enjoyment of such rights—for example, a regulation that recognizes the specificities of a child's artistic work in digital. Thirdly, this work led us to the hypothesis that the consumption logics that organize children's experience in the digital environment reconfigure children's participation practices in marketing practices, making them vehicles and targets of commercial discourses. These notes carry both potentialities and limitations of this work that can be explored and overcome, respectively, in future works.

**Author Contributions:** Conceptualization, R.T., B.G. and I.M.; methodology, R.T., B.G. and I.M.; formal analysis, R.T., B.G. and I.M.; investigation, R.T., B.G. and I.M.; resources, R.T., B.G. and I.M.; data curation, R.T., B.G. and I.M.; writing—review and editing, R.T., B.G. and I.M.; supervision, R.T. All authors have read and agreed to the published version of the manuscript.

**Funding:** This research received no external funding.

**Acknowledgments:** This work recognizes the Ética na Sociedade do Consumo (Ethics in the Consumer Society) Research Group—linked to the Federal Fluminense University, in Niteroi-RJ, Brazil—leaded by the professors Ana Paula Bragaglia and Patrícia Burrowes, as an important vector of convergence in our personal trajectories, and promoter of the studies and reflections that instigated (and continue instigating) our research.

**Conflicts of Interest:** The authors declare no conflict of interest.

## Notes

[1]   A complex perspective that presents consumption as culture, that is, as a formatting bias of our preferences, behaviors, relationships and even our self-perception in relation to the world. A culture whose main character is "the consumer", whose participation is scheduled and foreseen by the dynamics of the market, which, in turn, ratifies itself as a legitimate choice of the consumer (Fontenelle 2017).

[2]   Considering the existence of Children's Consumption Cultures presupposes understanding commoditization as a fundamental basis of contemporary children's cultures. A monumental achievement of 20th century capitalism, which intensified the market for children's goods, positioning it as a standard of current commercial practices, through dynamics that point to the framing of childhood itself as a commodity and as a central aspect of the flow of reproduction and transformation of Consumer Culture in general (Cook 2004).

[3]   The experience of child digital citizenship assumes specific conditions for the enjoyment of the rights safeguarded for children today, and takes into account criteria that include the qualitative provision of resources for the full use of the possibilities of effective participation of these subjects, on equal terms with adults, while benefiting in these processes from protective measures that consider their particular phase of development (UN 2021). In this sense, digital competences—such as knowledge, skills and attitudes—are required from users so that they can exercise their full citizenship by engaging with information and communication technologies in a confident, critical and safe way (Vuorikari et al. 2022).

[4]   Although Brazilian law grants the guarantees provided by the 1989 Convention to all persons under 18 years of age, it makes a distinction between children and adolescents. Children are between zero to 12 years old. Adolescents are between 12 and 18 years old. In this text, we do not adopt this distinction for children, a term by which we refer to all individuals protected by the Convention.

[5]   Available online: https://tinyurl.com/mryfmc7e, accessed on 7 December 2022.

[6]   Available online: https://tinyurl.com/bdf3rwhn, accessed on 7 December 2022.

[7]   Almost 40 thousand american dollars in 2020.

[8]   A project through which the State opts for minimal intervention in the economy, through its withdrawal from the market, which, in theory, regulates itself as well as the economic order. It is our understanding that advertising narratives and practices, as one of the main spokespersons of capital before the public, reinforce the perspective of this sort of society project.

[9]   Available online: https://tinyurl.com/5fts226f, accessed on 25 December 2022.

[10]  Figures from July 2022.

[11]  Available online: https://tinyurl.com/yen5p5cb, accessed on 19 September 2021.

[12]  Available online: https://tinyurl.com/2xkm733u, accessed on 19 September 2021.

[13]  Available online: https://tinyurl.com/3kua4zkw, accessed on 25 December 2022.

[14]  Available online: https://www.instagram.com/pietraquintela/, accessed on 25 December 2022.

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
