# Peer review of "Main Challenges for Child Digital Citizenship in a Consumer Culture in Brazil"

_journalmedia, doi:10.3390/journalmedia4010004_

Round 1

Reviewer 1 Report

The manuscript „Main Challenges for Child Digital Citizenship: The Rights of Brazilian Children Facing the Dynamics of Consumer Culture” is a theoretical contribution.

The topic is relevant to the target journal and, overall, the manuscript is readable. There still is room for improvement in several respects.

1)    The message of the paper title could be more on point

2)    The abstract lacks a conclusion

3)    Several main concepts are not sufficiently or not at all defined such as “child digital citizenship”, “child”, “consumer culture”, “rights-by-design”, “neoliberal discourses”

4)    Introduction is very long and lacks sub-headings

5)    Research aim should spelled out more clearly and earlier in the text

6)    A cultural contextualization is missing: How is the situation in Brazil similar or different to the situation in North America or Europe?

7)    Discussion section is missing and puts results in perspective and provides a critical self-reflection of the limitations of the theoretical paper

8)    A figure with a conceptual framework would be helpful

9)    A lot of non-English references are used that need a translation of the titles to English for international readers

10)                   No clear practical implications are outlined.

Author Response

Dear reviewer,
Thank you for your careful reading and recommendations. Please see the attached comments and responses.

Reviewer 2 Report

Thank you for the opportunity to review this paper. I found it both topical and interesting. However, a few points needs to be addressed:

-         No definition of digital citizenship is provided – the authors take a human rights approach to the concept, but can they put that in context in ways that acknowledge different approaches to digital citizenship (e.g., in terms of users’ ethical know-how and moral behaviour, or in terms of social participation)? I suggest the authors define the term as it first appears and then add a section or a paragraph focusing on digital citizenship, particularly in relation to children’s engagement with digital technologies. This is a key concept in the paper and, as it is a contested notion, it’d be good to have a section that acknowledges work in this area and explains how the authors approach the term.

-          Section 4 – how did the authors analyse the studies? Was this a literature review? Which studies were selected and how? How did the authors synthetise the material so as to generate the themes that they present in section 4? More clarity is needed here as to the process behind the selection and analysis of the studies chosen by the authors. If the lit review was not systematic (that is, based on a developed and tested search strategy used to find article across different databases), then the authors might want to acknowledge this as a limitation of their study.

-          The authors might want add a section at the end reflecting on the implications of their arguments for digital citizenship, particularly in the context of education – that is, what can we learn from their paper in relation to how digital citizenship education could be better promoted through education?

-          Similarly, the authors should reflect on the limitations of their study, while also considering what future directions for both research and practice might follow.

Author Response

Dear reviewer,
Thank you for your careful reading and recommendations. Please see the attached comments and responses.

Yours sincerely,
Authors

Round 2

Reviewer 1 Report

The revision and answer letter have sufficiently addressed my concerns